# Prevalence and predictors of Biker's hand syndrome among the professional bike riders in Bangladesh: A cross-sectional study

Sohel Ahmed[1,2]*, Mohammad Jahirul Islam[3], G. M. Jakaria[4], Md Enamul Haque[5], Jalal Uddin[6], Tazveen Fariha[7], Md Saifur Rahman[8], Md Zahidul Islam[9], Raju Ahmed[10], Selim Hossain[11], S. M. Mahfuz Anwar[12]

**1** Ahmed Physiotherapy & Research Center, Dhaka, Bangladesh, **2** Directorate of Students' Welfare, Bangladesh University of Engineering and Technology, Dhaka, Bangladesh, **3** Department of Physical Medicine and Rehabilitation, MAG Osmani Medical College Hospital, Sylhet, Bangladesh, **4** Department of Physiotherapy, Lincoln University College, Selangor Darul Ehsan, Malaysia, **5** Department of Public health, North south University, Dhaka, Bangladesh, **6** Department of Epidemiology and Biostatistics, School of Public Health, University of Nevada Las Vegas, Maryland Parkway, Las Vegas - NV, United States of America, **7** Department of Social and Behavioral Health, School of Public Health, University of Nevada Las Vegas, Maryland Parkway, Las Vegas-NV, United States of America, **8** Department of Physiotherapy, Saic College of Medical Science and Technology, Dhaka, Bangladesh, **9** Department of Physiotherapy, Dhaka College of Physiotherapy, Dhaka, Bangladesh, **10** Department of Physiotherapy, Gonoshasthay Samaj Vittik College of Physiotherapy and Health Science, Dhaka, Bangladesh, **11** Department of Orthopedic, Sheikh Hasina Medical College, Habiganj, Bangladesh **12** Department of Orthopaedic, Habiganj Medical College, Habiganj, Bangladesh

* ptsohel@gmail.com

## Abstract

### Background

The use of motorbikes for the rapid growth of ride-sharing platforms in Bangladesh and the increasing economic reliance on this occupation are significant sources of non-traumatic injuries, particularly those affecting the wrist and hand. This study aimed to determine the prevalence and factors that contribute to the occurrence of biker's hand syndrome among bike riders in Bangladesh.

### Methods

A cross-sectional study was conducted among professional bike riders using the Cornell hand discomfort questionnaire. This study includes 630 male bike riders from the cities of Dhaka and Sylhet in Bangladesh, selected through two-stage cluster sampling using a face-to-face interviewer-administered questionnaire. A binary logistic regression analysis was used to identify the factor that predicts biker's hand syndrome.

### Results

The prevalence of pain, ache, or discomfort was reported as 58.3% in the right hand (RH) and 51.3% in the left hand (LH). Participants who ride bikes with an

**Data availability statement:** All relevant data are within the manuscript and its supporting information files.

**Funding:** The author(s) received no specific funding for this work.

**Competing interests:** The authors have declared that no competing interests exist.

engine capacity of less than 150cc (aOR 2.218, CI 1.192–4.128, p = 0.012 in RH and aOR 1.210, CI 0.672–2.157, p = 0.525 in LH), and uncomfortable handlebar (aOR 2.110, CI 1.171–3.801, p = 0.013 in RH and aOR 1.519, CI 0.888–2.598, p = 0.127 in LH) reported higher likelihood of hand syndrome. Individuals whose motorcycle does not fit their body physique have over five times higher odds of hand discomfort (aOR 5.136, CI 2.939–8.974, p < 0.001 in RH and aOR 3.676, CI 2.210–6.113, p < 0.001 in LH).

## Conclusion

This study highlights a considerably greater occurrence of hand syndrome among professional bikers in Bangladesh. Implementing regular stretching exercises, the use of a comfortable hand gripper, and improving motorcycle-to-body fit could reduce hand syndrome risk by over fivefold. Policymakers and organizations involved in occupational health and safety should priorities taking appropriate measures to address this issue.

## Introduction

Ride-sharing through motorbike is gaining popularity in Bangladesh [1] because of their affordability and flexibility, and considered one of the most popular modes of transportation [2]. From 2017 ride sharing through motorbike has begun and Pathao and Uber gaining popularity in Bangladesh [3]. Work-related musculoskeletal diseases are a prevalent set of occupational disorders in which job exposures have a significant causative influence. Motorbike riding is a multifaceted activity that requires riders to maintain balance, focus their attention, and engage in physical exertion to maintain control. Moreover, the presence of hand vibrations [4], prolonged working hours, poor ergonomic conditions, intricate design of the motorcycle, and the act of transporting people all contribute to the increased difficulty faced by motorcyclists [5].

Biker's hand syndrome is a condition characterized by pain, discomfort, paresthesia, or numbness in the hands. The symptoms result from excessive usage, nerve compression, or excessive pressure on the hand during prolonged durations of bike riding [6–11]. Although motorcycles are heavier, produce more vibration, operate at higher speeds, and are often used for longer periods than mountain biking and road cycling, these symptoms are prevalent in mountain biking [10] and road cycling [8], as shown in prior research. Hand injury is a common non-traumatic injury associated with professional biking reported in a cross-sectional study [12] and cycling reported in a literature review study [6].

Occupational bike riders have a range of physical challenges, such as extensive bike riding, hand vibrations, and prolonged working hours in awkward posture, all of which may lead to various musculoskeletal pain [12] and hand discomfort [13,14]. Hand/Arm symptoms cause productivity loss, which includes 11% sickness absence and 36% productivity loss [15], eventually resulting in a significant burden

on healthcare resources [14,16]. Hence, it is worthwhile to examine the prevalence and contributing factors of biker hand syndrome among professional bike riders. Previous studies have only examined pain and discomfort in the hand as a component of research on musculoskeletal disorders [13,14] or hand-arm vibration syndrome [4]. Risk factors may encompass individual characteristics such as age and BMI, occupational exposures including riding duration and handle-bar design, as well as biomechanical stressors like vibration and sustained grip force, which collectively contribute to the development of hand symptoms. To the best of our knowledge, no prior study has specifically investigated the prevalence and predictors of hand discomfort among professional ride-sharing motorcycle riders in the context of Bangladesh. This study seeks to ascertain the prevalence of biker's hand syndrome and to identify modifiable risk factors among professional ride-sharing motorcycle riders in Bangladesh, aiming to inform preventive measures in occupational health.

## Methods

### Study design

This cross-sectional research was done among professional bike riders in the Dhaka and Sylhet metropolitan regions of Bangladesh from January to June 2024. The term "professional biker" specifically refers to ride-sharing drivers (such as Pathao and Uber) who generate income by providing ride-sharing services and who ride bikes for at least 3–4 hours each day.

### Ethical consideration

The Institute of Physiotherapy, Rehabilitation, and Research, which is affiliated with the Bangladesh Physiotherapy Association, received permission from the ethical review board. The clearance was obtained under the reference number BPA-IPRR/IRB/07/12/2023/119. This study, which included human subjects, strictly followed the ethical principles outlined by the Bangladesh Medical Research Council and the Declaration of Helsinki (2013 revised). Prior the participants took part; we provided a brief summary of the study's nature and purposes and solicited their voluntary participation. We assure participants that they may withdraw from the study at any time without penalty. Furthermore, their information will be kept confidential and will only be used for publication in the journal. We acquired digitally informed consent from the participants by selecting either the consent or decline response in the Google form prior to their involvement in this research.

### Study population

Professional bike riders from Dhaka and Sylhet metropolitan city served as the population source. The participants in this research were individuals who offer passenger transport via different ride-sharing applications like Uber, Pathao etc., and only those who satisfied the inclusion and exclusion criteria were considered.

### Sampling technique and data collection

Data were collected from Dhaka and Sylhet metropolitan cities in Bangladesh, using a dual-stage cluster sampling method to include potential participants. To minimize selection bias and ensure a diverse population, we randomly selected bus and train stations, markets, and other public areas where we could find bikers, which were then grouped together as clusters. For cluster randomization, we began by identifying the potential bus and train stations in Dhaka and Sylhet. Initially, we gathered data from all the intercity bus and train stations, which comprised three train stations (one in Sylhet and two in Dhaka) and five bus stations (one in Sylhet and four in Dhaka). Following this, we created a list of the local bus stations and markets and selected twenty potential locations using a lottery method. As we were unable to ascertain an exact number of bike riders reliant on their expenditures for biking, we selected individuals randomly who were there during data collection time and provided consent to participate. This method facilitated the inclusion of persons from diverse locations. In order to prevent duplicate collection of data for the same bikers, we included the bike registration number as a

unique identifying ID. The survey was conducted using an interviewer-administered questionnaire. Six data collectors, all of whom had graduate degrees in physiotherapy and health sciences, were selected for the purpose of data collection. An interviewer-administered questionnaire was utilized to gather data. This questionnaire underwent translation into Bangla and was subsequently translated back into English with the help of experts. Before data collection began, all interviewers participated in a training session facilitated by the primary researcher to ensure consistency in their approach with different interviewees. The data collector initially posed a question to the responder, and after receiving an answer, the data collector repeated the response to confirm its accuracy.

## Subject selection criteria

**Inclusion criteria.** This study included professional bike riders who offer passenger transport via different ride-sharing applications between the ages of 18 and 50 who are registered members of any riding service and are willing to participate.

**Exclusion criteria.** This study excluded participants who had any documented history of hand surgeries or road traffic accidents in the last six months. We created a checklist with yes-or-no questions prior to data collection to ensure compliance with these requirements. We also excluded food or other delivery riders from this study.

## Sample size estimation

The sample size was obtained using the approach for single population proportions, assuming a 95% level of significance, a 4% margin of error, and a response distribution of 50% [17]. The sample size computation used the following variables: n for sample size, p for prevalence, and d for margin of error.

$$N = z^2 \, p \, (1-p)/ \, d^2, \ n \ = \ (1.96)^2 \times (0.5) \, (0.5)/ \, (0.04)^2, \ = \ 600$$

The ultimate sample size consisted of 630 samples, which included 5% incomplete forms.

## Survey development

***Socio-demographic variables.*** We included demographic characteristics such as age, weight, height, BMI, educational qualification (SSC or bellow, HSC, Under-graduate, Post-graduate), marital status (Married and Unmarried/ Divorced/Widow), gross monthly income (≤30000, 31000–50000, and >50000 Taka), smoking (Yes/No), and alcohol habit (Yes/No). This study also includes work-related factors such as the rider's status (either a professional who generates income solely from ride sharing or an occasional rider who shares rides for 3–4 hours to earn extra income alongside other work), average riding hours per day (<6 hours, 6–8 hours, > 8 hours), and year of riding experience (0–5 years, 6–10 years, > 10 years). We also asked a few questions, such as: Is the hand grip on your bike comfortable for you (Yes/No)? Do you think your bike fits according to your body physique (Yes/No)? Do you use any safety equipment while riding the bike (Yes/No)? and bike engine capacity (≤150 cc/ > 250 cc) were included.

**Cornell Hand Discomfort Questionnaire (CHDQ).** The CHDQ includes a hand map diagram with six shaded areas (Fig 1) representing different parts of the hand. It is used to assess the frequency of musculoskeletal pain, ache or discomfort, it's intensity and how its interference at work in the last working week. In this research we utilized only the presence or absence of pain, ache or discomfort in different shaded area of the hands for last seven days using this questionnaire. Dr. Oguzhan Erdinc conducted an assessment of the CMDQ's validity and obtained favorable results [18].

## Survey administration

Each interviewer first tested the final draft of the questionnaire on five representative populations to gauge response times and familiarity with data collection; the average time was less than 10 minutes. The questionnaire worked well for both the

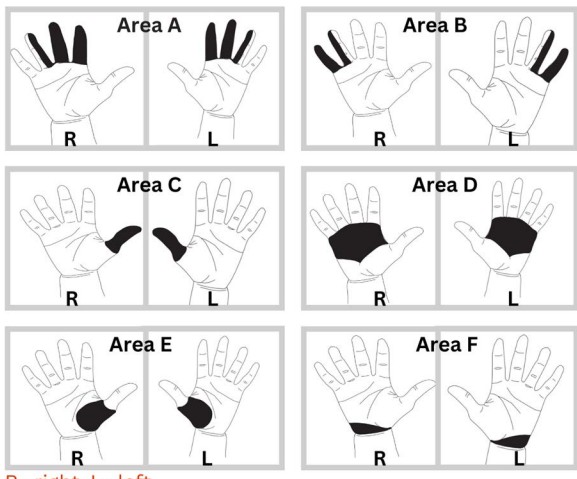

R= right, L= left

**Fig 1. Hand map diagram of six shaded area of hand.**

data collector and the participants, and no modifications were required. We used the Google Forms® platform to gather responses to the survey as it is eco-friendly and it's built in require option prevent incomplete submission. The data collector conducted a face-to-face survey by utilizing the Google Forms platform on a mobile phone or tab. Every question on Google Forms includes the necessary option to prevent incomplete submissions. We conveniently invited approximately 750 participants to participate in individualized interview sessions using a semi-structured questionnaire. We collected data from 630 participants as a result of a 16% refusal rate. The main reasons for their refusal in the study were insufficient time and an unwillingness to engage. The final analysis ultimately included 630 respondents. The blueprint of the trial is reported in Fig 2.

## Measurement of variables

**Independent variables.** We categorized age into three distinct groups: 18–30 years, 31–40 years, and above 40 years. In addition, we classify additional factors into the following categories: The variables to consider are Body Mass Index (BMI) categorized as underweight (<18.5 kg/m²), normal (18.5–22.9 kg/m²), or overweight (23.0–24.9 kg/m²) and obesity (>25.0 kg/m²); level of education categorized as secondary school certificate (SSC) or below, higher-secondary school certificate (HSC), or Graduate and above; marital status categorized as married, and unmarried or divorced, or widow; monthly family income categorized as 15000 taka, 15000–30000 taka, 31000–45000 taka, or above 45000 taka; rider status categorized as professional or occasional; smoking categorized as no or yes; riding time categorized as less than 6 hours, 6–8 hours, or more than 8 hours; bike engine capacity categorized as ≤150cc, or above 150cc; use of handle gripper categorized as yes or no; use of safety equipment categorized as yes or no; and whether the bike fits your body's physique categorized as yes or no.

## Dependent variables

**Cornell Hand Discomfort Questionnaire.** Initially, we gathered binary answers indicating the presence or absence of pain in both hands across six shaded areas. If any shaded part of the hand exhibited pain, we interpreted it as a yes response; conversely, if there was no pain, we took it as a no response as a binary variable for both hands.

## Data analysis

Descriptive statistics were used to represent data using measures such as frequency, percentage, mean, and standard deviation. The Chi-square test was used to examine any possible correlation between hand pain, aching, or discomfort

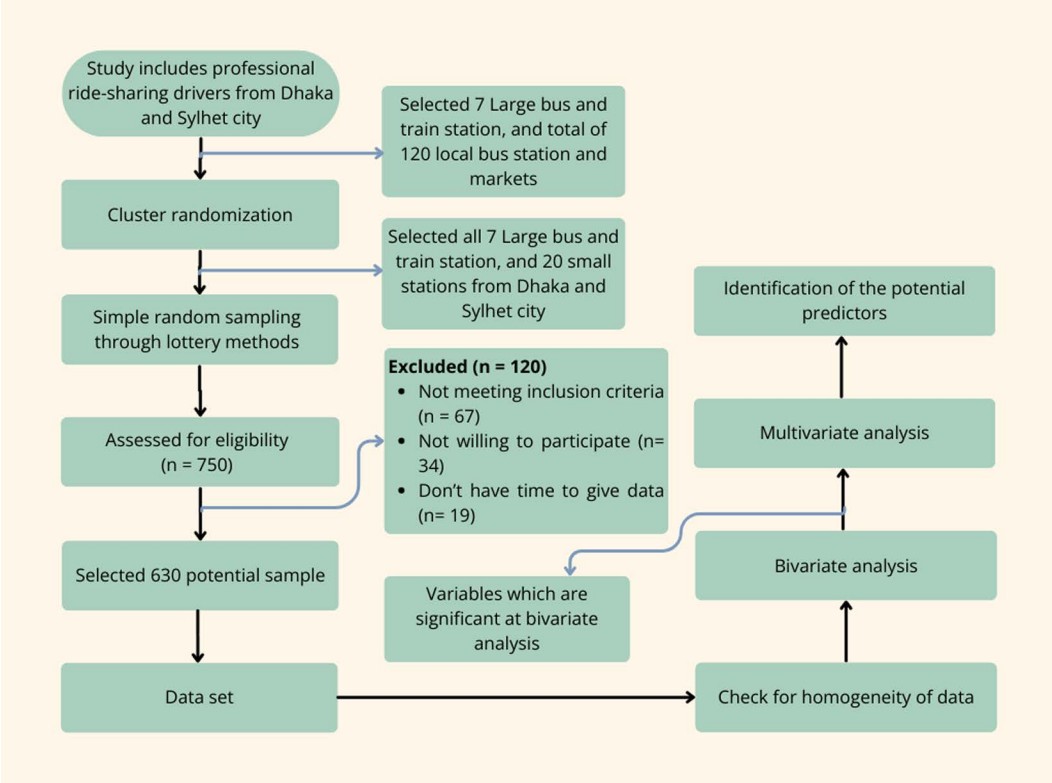

**Fig 2. Flow chart summarizing the study procedure.**

and socio-demographic factors. Binary logistic regression analysis (enter method) was employed using odds ratios (OR) and a 95% confidence interval (CI) to determine the factor that predicts for biker's hand syndrome. The model's adequacy was determined by using Hosmer-Lemeshow's test and a categorization table. If the p value is greater than 0.05, we consider the model to be a good fit for logistic regression. Variance inflation factors (VIFs) were used to assess the presence of multicollinearity among the independent variables. A threshold of VIF ≤ 3.0 was applied. The threshold of significance for each test was set at a value <0.05. The final model showed good fit (Hosmer-Lemeshow $\chi^2 = 4.765$, p = 0.782 for the right hand and $\chi^2 = 10.68$, p = 0.220 for the left hand), and all VIF values were below 3, indicating no multicollinearity. The statistical analysis was conducted with IBM SPSS Version 27.

## Results

### Socio-demographic characteristics of the participants

The current research consisted of a total of 630 male participants as all forms were complete with no missing data. Since there were no forms that were not completed, the final analysis included all of the samples. The mean age, weight, height, and BMI of the participants were 33.73 ± 7.12 years, 66.57 ± 8.51 kg, 166.86 ± 5.53 cm, and 23.95 ± 3.10 kg/m2, respectively. Approximately 44.2% of the individuals successfully finished their secondary school, while 74.4% were in a state of matrimony. The majority of the participants (75.1%) were professional ride-sharing riders, and over two-thirds (68.1%) had less than five years of riding experience. More than half of the participants (67.9%) had a monthly income of <30000 Bangladeshi taka and 39.7% of them engaged in riding for 6–8 hours daily. Maximum participants (85.7%) ride less engine

capacity bike (≤150 cc) and 81.4% reported their handlebar and 77.3% reported their bike is suit-fit to their body. The information is provided in Table 1.

## Prevalence of hand syndrome among bikers

In this research, 58.3% of participants had pain, aching, or discomfort in any of the six shaded parts of their right hand. Among them, 20.2% reported experiencing these symptoms 1–2 times per day, 10.3% reported 3–4 times per day, 18.1% reported once a day, and 6.7% reported experiencing them numerous times per day. 30.8% of individuals had mild discomfort, 14.4% experienced moderate discomfort, and 6.8% experienced severe discomfort. 24.3% of participants had discomfort that had a little impact on their job, while 8.3% said that the pain had a significant impact on their work. On the other hand, 51.3% of individuals had pain or discomfort in their left hand. Among them, 19.8% reported experiencing these symptoms 1–2 times per day, 10.6% reported 3–4 times per day, 14.0% reported experiencing them once a day, and 2.9% reported experiencing them numerous times per day. 33.0% of individuals had mild discomfort, 8.4% experienced moderate discomfort, and 2.9% experienced severe discomfort. 23.8% of participants reported discomfort that had little impact on their job, while 3.8% said that the pain had a significant impact on their work. The prevalence of pain in different areas of the right hand was as follows: 20.5% of participants reported pain in area A, 12.7% in area B, 16.5% in area C, 13.0% in area D, 16.8% in area E, and 22.2% in area F region. In the left hand, area A had the highest incidence of pain, with a rate of 16.2%. This was followed by region B at 10.5%, area C at 10.2%, area D at 16.7%, area E at 10.5%, and area F at 14.3% Table 2.

## Association between hand pain, ache or discomfort and socio-demographic variables

Hand discomfort were significantly associated with the participants' age (p < 0.001 right and p < 0.001 left), BMI (p < 0.001 right and p < 0.001 left), education (p < 0.001 right and p = 0.003 left), and marital status (p < 0.001 right and p < 0.001 left). There was also a significant association between gross monthly income (p < 0.001 right and p < 0.001 left), rider status (p = 0.002 right and p < 0.001 left), average riding time in a day (p < 0.001 right and p < 0.001 left), experience as a bike rider (p < 0.001 right and p < 0.001 left), fittingness of the bike according to the body (p < 0.001 right and p < 0.001 left) and hand grip suitability (p = 0.001 for right and p < 0.001 for left). Bike engine capacity was only significant for right hand (p = 0.002). While no significant association was observed with the smoking habit (p = 0.094 right and p = 0.211 left) and the use of safety equipment (p = 0.827 right and p = 0.281 left).

## Predictors of bikers' hand syndrome

Compared to participants over 40 years of age, those aged 31–40 (aOR 0.267, CI 0.113–0.534, p < 0.001 for the right hand and aOR 0.332, CI 0.172–0.637, p = 0.001 for the left hand) had significantly lower odds of hand syndrome Participants with BMI < 18.5 kg/m² (aOR 1.162, CI 0.681–1.985, p = 0.582 in RH and aOR 1.583, CI 0.940–2.667, p = 0.084 in LH) and BMI 18.5–22.9 kg/m² (aOR 2.010, CI 1.238–3.259, p = 0.005 in RH and aOR 1.628, CI 1.015–2.610, p = 0.043 in LH) reported less pain compared with overweight participants.. Participants with lower educational qualifications, specifically HSC (aOR 2.929, CI 1.558–5.506, p = 0.001 for the right hand and aOR 1.791, CI 0.982–3.268, p = 0.057 for the left hand), and ≤SSC (aOR 1.867, CI 1.188–2.933, p = 0.007 for the right hand and aOR 1.001, CI 0.644–1.557, p = 0.995 for the left hand) exhibited a higher susceptibility to hand syndrome compared to those with higher educational qualifications (bachelor's degree and above). Additionally, individuals with 0–5 years of experience as bike riders (aOR 2.062, CI 0.823–5.167, p = 0.112 for the right hand and aOR 3.472, CI 1.511–7.977, p = 0.003 for the left hand) demonstrated an increased prevalence of hand syndrome. Additionally, increased adjusted odds of hand syndrome is observed among participants who rides bikes whose engine capacity is ≤ 150cc (aOR 2.218, CI 1.192–4.128, p = 0.012 for right hand and aOR 1.210, CI 0.672–2.157, p = 0.525 for left hand), hand grip for the bike is not suitable (aOR 2.110, CI 1.171–3.801, p = 0.013 for right hand and aOR 1.519, CI 0.888–2.598, p = 0.127 for left hand) and who thought their bike is not fitted according to their

**Table 1. Socio-demographic characteristics of the study participants.**

| Variables | Number | Percentage |
|---|---|---|
| **Age** | | |
| 18-30 years | 231 | 36.7 |
| 31-40 years | 288 | 45.7 |
| >40 years | 111 | 17.6 |
| **BMI** | | |
| <18.5 kg/m² | 12 | 1.9 |
| 18.5-22.9 kg/m² | 230 | 36.5 |
| 23.0-24.9 kg/m² | 145 | 23.0 |
| ≥ 25 kg/m² | 243 | 38.6 |
| **Educational qualification** | | |
| ≤ SSC | 227 | 36.0 |
| HSC | 278 | 44.2 |
| Graduation and above | 125 | 19.8 |
| **Marital status** | | |
| Married | 469 | 74.4 |
| Unmarried/Divorced/Widow | 161 | 25.6 |
| **Smoking habit** | | |
| Yes | 421 | 66.8 |
| No | 209 | 33.2 |
| **Gross monthly income** | | |
| ≤ 30000 | 428 | 67.9 |
| 31000-45000 | 133 | 21.1 |
| >45000 | 69 | 11.0 |
| **Rider status** | | |
| Occasional | 157 | 24.9 |
| Professional | 473 | 75.1 |
| **Average riding time in a day** | | |
| < 6 hours | 283 | 44.9 |
| hours | 250 | 39.7 |
| > 8 hours | 97 | 15.4 |
| **Experience as a rider** | | |
| 0-5 years | 429 | 68.1 |
| 6-10 years | 143 | 22.7 |
| >10 years | 58 | 9.2 |
| **Bike engine capacity** | | |
| ≤150 cc | 540 | 85.7 |
| > 150 cc | 90 | 14.3 |
| **Is the hand grip of your bike suitable for you?** | | |
| Yes | 513 | 81.4 |
| No | 117 | 18.6 |
| **Do you think your bike fits according to your body?** | | |
| Yes | 487 | 77.3 |
| No | 143 | 22.7 |
| **Do you use safety equipment while riding the bike?** | | |
| Yes | 608 | 96.5 |
| No | 22 | 3.5 |

*(Continued)*

**Table 1.** (Continued)

| Variables | | Number | Percentage |
|---|---|---|---|
| Do you have pain in the shaded areas of the right hand? | | | |
| Area A | Yes | 129 | 20.5 |
| | No | 501 | 79.5 |
| Area B | Yes | 80 | 12.7 |
| | No | 550 | 87.3 |
| Area C | Yes | 104 | 16.5 |
| | No | 526 | 83.5 |
| Area D | Yes | 82 | 13.0 |
| | No | 548 | 87.0 |
| Area E | Yes | 106 | 16.8 |
| | No | 524 | 83.2 |
| Area F | Yes | 140 | 22.2 |
| | No | 490 | 77.8 |
| During the last work week how often did you experience ache, pain and discomfort in your Right hand | | | |
| Never | | 282 | 44.8 |
| 1-2 times | | 127 | 20.2 |
| 3-4 times | | 65 | 10.3 |
| Once a day | | 114 | 18.1 |
| Multiple times a day | | 42 | 6.7 |
| If you experienced ache, pain discomfort, how uncomfortable was this in your right hand? | | | |
| Not experienced pain | | 302 | 47.9 |
| Slightly uncomfortable | | 196 | 30.8 |
| Moderately uncomfortable | | 91 | 14.4 |
| Very uncomfortable | | 43 | 6.8 |
| If you experienced ache, pain discomfort, did this interfere your work in right hand? | | | |
| Not experienced pain | | 265 | 42.1 |
| Not at all | | 160 | 25.4 |
| Slightly interfered | | 153 | 24.3 |
| Substantially interfered | | 52 | 8.3 |
| Do you have pain in the shaded areas of the left hand? | | | |
| Area A | Yes | 102 | 16.2 |
| | No | 528 | 83.8 |
| Area B | Yes | 66 | 10.5 |
| | No | 564 | 89.5 |
| Area C | Yes | 64 | 10.2 |
| | No | 566 | 89.8 |
| Area D | Yes | 105 | 16.7 |
| | No | 525 | 83.3 |
| Area E | Yes | 66 | 10.5 |
| | No | 564 | 89.5 |
| Area F | Yes | 90 | 14.3 |
| | No | 540 | 85.7 |
| During the last work week how often did you experience ache, pain and discomfort in your left hand | | | |
| Never | | 332 | 52.7 |
| 1-2 times | | 125 | 19.8 |

*(Continued)*

**Table 1.** (Continued)

| Variables | Number | Percentage |
|---|---|---|
| 3-4 times | 67 | 10.6 |
| Once a day | 88 | 14.0 |
| Multiple times a day | 18 | 2.9 |
| If you experienced ache, pain discomfort, how uncomfortable was this in your left hand? | | |
| Not experienced pain | 350 | 55.6 |
| Slightly uncomfortable | 208 | 33.0 |
| Moderately uncomfortable | 53 | 8.4 |
| Very uncomfortable | 19 | 3.0 |
| If you experienced ache, pain discomfort, did this interfere your work in left hand? | | |
| Not experienced pain | 290 | 46.0 |
| Not at all | 166 | 26.3 |
| Slightly interfered | 150 | 23.8 |
| Substantially interfered | 24 | 3.8 |

body physique (aOR 5.136, CI 2.939–8.974, p<0.001 for right hand and aOR 3.676, CI 2.210–6.113, p<0.001 for left hand) See (Table 3) for more details.

## Discussion

The objective of this cross-sectional investigation was to determine the prevalence and predictors of hand syndrome among professional bike riders in Bangladesh. This investigation demonstrated that the prevalence of hand discomfort in any of the six shaded regions of the right hand is approximately 58.3%, while the prevalence in the left hand is approximately 51.3%. To the best of our knowledge, no prior study has investigated the prevalence of hand syndrome among professional bike riders. A cross-sectional study investigated musculoskeletal pain and discomfort among adult male motorcyclists. 25% of motorcyclists complained of pain in their wrist and hand [12]. Another study investigating musculoskeletal disorders among takeaway riders in China reported that the prevalence of wrist and hand pain is about 9.1% [14]. In a cross-sectional study, the prevalence of hand and wrist pain among three-wheeler car drivers in Ethiopia was reported to be around 35.5% [19].

The study revealed that hand pain was significantly associated with a variety of socio-demographic and occupational factors. Individuals aged less than 40 had a lower prevalence of hand syndrome than those over 40. In the present study, participants with the BMI normal weight developed hand syndrome at twice the rate for the right hand and one and a half times for the left hand and underweight participants demonstrated a higher odd of being developed hand syndrome compared to the obese participants. While driving a motorcycle, controlling and handling it is a difficult task. Low-BMI (underweight or normal weight) riders may have trouble handling the motorcycle due to less physical capability leads to develop hand syndrome. However, obese individuals in this group may have lower exposure to hand stressors, such as fewer riding hours or different motorcycle types, resulting in the development of less pain syndrome. It also might be due to the misclassification, confounding or statistical artifact. A cross-sectional study reported that backpacks weighing more than 10% of body weight had twice the risk of musculoskeletal pain, [20] which suggests that working capacity and loading should be optimal to get rid of musculoskeletal pain.

Bikers with a monthly income exceeding 45,000 Bangladeshi Taka experienced hand syndrome at a rate approximately one and a half times higher than those with a lower income. The findings are similar to previous published literature reporting a significant association between musculoskeletal disorders and monthly income among takeaway drivers in China [14]. This study also found a significant association between years of professional experience and hand syndrome

**Table 2. Association between hand pain, ache or discomfort and socio-demographic variables of the participants.**

| Variables | Right hand | | | | | | Left hand | | | | |
|---|---|---|---|---|---|---|---|---|---|---|---|
| | Total (n%) | Yes (n%) | No (n%) | Degree of freedom | Chi-square statistics | p-value | Yes (n%) | No (n%) | Degree of freedom | Chi-square statistics | p-value |
| Age | | | | | | | | | | | |
| 18-30 years | 231(36.7) | 95(41.1) | 136(58.9) | 2 | 50.59 | <0.001* | 77(33.3) | 154(66.7) | 2 | 51.54 | <0.001* |
| 31-40 years | 288(45.7) | 185(64.2) | 103(35.8) | | | | 168(58.3) | 120(42.7) | | | |
| >40 years | 111(17.6) | 87(78.4) | 24(21.6) | | | | 78(70.3) | 33(29.7) | | | |
| BMI | | | | | | | | | | | |
| <18.5 kg/m2 | 12(1.9) | 04(33.3) | 08(66.7) | 3 | 18.84 | <0.001* | 03(25.0) | 09(75.0) | 3 | 16.36 | 0.001* |
| 18.5-22.9 kg/m2 | 230(36.5) | 150(65.2) | 80(34.8) | | | | 122(53.0) | 108(47.0) | | | |
| 23.0-24.9 kg/m2 | 145(23.0) | 94(64.8) | 51(35.2) | | | | 91(62.8) | 54(37.2) | | | |
| ≥ 25 kg/m2 | 243(38.6) | 119(49.0) | 124(51.0) | | | | 107(44.0) | 136(56.0) | | | |
| Educational Qualification | | | | | | | | | | | |
| ≤ SSC | 227(36.0) | 110(48.5) | 117(51.5) | 2 | 21.20 | <0.001* | 109(48.0) | 118(52.0) | 2 | 11.43 | 0.003* |
| HSC | 278(44.1) | 165(59.4) | 113(40.6) | | | | 133(47.8) | 145(52.2) | | | |
| Graduation and above | 125(19.8) | 92(73.6) | 33(26.4) | | | | 81(64.8) | 44(35.2) | | | |
| Marital status | | | | | | | | | | | |
| Married | 469(74.4) | 295(62.9) | 174(37.1) | 1 | 16.29 | <0.001* | 266(56.7) | 203(43.3) | 1 | 21.79 | <0.001* |
| Unmarried/Divorced/Widow | 161(25.6) | 72(44.7) | 89(55.3) | | | | 57(35.4) | 104(64.6) | | | |
| Smoking habit | | | | | | | | | | | |
| Yes | 421(66.8) | 255(60.6) | 166(39.4) | 1 | 2.80 | 0.094 | 225(53.4) | 196(46.6) | 1 | 2.40 | 0.121 |
| No | 209(33.2) | 112(53.6) | 97(46.4) | | | | 98(46.9) | 111(53.1) | | | |
| Gross monthly income in Bangladeshi Taka | | | | | | | | | | | |
| ≤ 30000 | 428(67.9) | 208(48.6) | 220(51.4) | 2 | 51.19 | <0.001* | 177(41.4) | 251(58.6) | 2 | 52.59 | <0.001* |
| 31000-45000 | 133(21.1) | 105(78.9) | 28(22.1) | | | | 97(72.9) | 36(27.1) | | | |
| >45000 | 69(11.0) | 54(78.3) | 15(21.7) | | | | 49(71.0) | 20(29.0) | | | |
| Rider status | | | | | | | | | | | |
| Occasional | 157(24.9) | 75(47.8) | 82(52.2) | 1 | 9.45 | 0.002* | 53(33.8) | 104(66.2) | 1 | 25.67 | <0.001* |
| Professional | 473(75.1) | 292(61.7) | 75(47.8) | | | | 270(57.1) | 203(42.9) | | | |
| Average riding time in a day | | | | | | | | | | | |
| < 6 hours | 283(44.9) | 131(46.3) | 152(53.7) | 2 | 35.04 | <0.001* | 102(36.0) | 181(64.0) | 2 | 50.67 | <0.001* |
| Hours | 250(39.7) | 161(64.4) | 89(35.6) | | | | 152(60.8) | 98(39.2) | | | |
| > 8 hours | 97(15.4) | 75(77.3) | 22(22.7) | | | | 69(71.1) | 28(28.9) | | | |
| Experience as a bike rider | | | | | | | | | | | |
| 0-5 years | 429(68.1) | 206(48.0) | 223(52.0) | 2 | 58.53 | <0.001* | 175(40.8) | 254(59.2) | 2 | 63.44 | <0.001* |
| 6-10 years | 143(22.7) | 117(81.8) | 26(18.2) | | | | 112(78.3) | 31(21.7) | | | |
| >10 years | 58(9.2) | 44(59.9) | 14(24.1) | | | | 36(62.1) | 22(37.9) | | | |
| Bike engine capacity | | | | | | | | | | | |
| ≤150 cc | 540(85.7) | 300(55.6) | 240(44.4) | 1 | 11.31 | 0.002* | 270(50.0) | 270(50.0) | 1 | 2.44 | 0.118 |
| > 150 cc | 90(14.3) | 67(74.4) | 23(35.6) | | | | 53(58.9) | 37(41.1) | | | |
| Is the hand grip of your bike suitable for you? | | | | | | | | | | | |
| Yes | 513(81.4) | 275(53.6) | 238(46.4) | 1 | 24.53 | <0.001* | 271(52.8) | 242(47.2) | 1 | 18.55 | <0.001* |
| No | 117(18.6) | 92(78.6) | 25(21.4) | | | | 81(69.2) | 36(30.8) | | | |

*(Continued)*

**Table 2.** (Continued)

| Variables | Right hand | | | | | | Left hand | | | | |
|---|---|---|---|---|---|---|---|---|---|---|---|
| | Total (n%) | Yes (n%) | No (n%) | Degree of freedom | Chi-square statistics | p-value | Yes (n%) | No (n%) | Degree of freedom | Chi-square statistics | p-value |
| Do you think your bike fits according to your body? | | | | | | | | | | | |
| Yes | 487(77.3) | 247(50.7) | 240(49.3) | 1 | 50.09 | <0.001* | 216(44.4) | 271(55.6) | 1 | 41.08 | <0.001* |
| No | 143(22.7) | 120(83.9) | 23(16.1) | | | | 107(74.8) | 36(25.2) | | | |
| Do you use safety equipment while riding the bike? | | | | | | | | | | | |
| Yes | 608(96.5) | 355(58.4) | 253(41.6) | 1 | 0.129 | 0.827 | 309(50.8) | 299(49.2) | 1 | 1.39 | 0.281 |
| No | 22(3.5) | 12(54.5) | 10(45.5) | | | | 14(63.6) | 08(36.4) | | | |

BMI – body mass index; SSC – secondary school certificate; HSC – higher secondary school certificate; * - p value significant at <0.05

among professional bikers. Riders with less than five years of experience are more likely to experience hand discomfort, as indicated by this study. This finding is clinically significant and may suggest inadequate ergonomic adaptation or improper riding techniques. Policymakers should address this concern by focusing on proper ergonomic design and developing targeted training programs for new riders. A study found that the duration of motorcycle riding and hand-arm vibration were significant risk factors for the development of musculoskeletal disorders among male traffic policemen who used high-powered motorcycles [13].

Furthermore, this study found increased hand syndrome among participants who rides less engine capacity bike. Ali et.al., reported in their study that riding less cubic capacity bike may trigger low back pain among ride-sharing bike drivers [5]. One may argue that smaller cubic capacity bikes are more rigid, uncomfortable, and challenging to maneuver than sports and higher cubic capacity bikes. Motorcyclists who utilize uncomfortable bikes for extended periods of time are likely to develop hand syndrome. The nature of riding a motorbike, including controlling and handling the vehicle, could account for the increased incidence of hand syndrome observed in this investigation. According to this study, hand syndrome is predicted by occupational characteristics such as cubic capacity and hand grip appropriateness in individuals.

Furthermore, the suitability of the hand grip and the bike's fit in relation to the bike rider's body shape and build were also significant predictors of the development of hand syndrome among professional bike riders in Bangladesh. A cross-sectional study conducted in Bangladesh among professional bike riders revealed that those who ride bikes with a displacement of less than 150cc have approximately 1.5 times higher odds of developing low back pain [5]. After bike riding, there was a significant increase in hand strength and discomfort, particularly in the left hand [21]. Motorbike riding is a complex task that requires riders to maintain balance and exert physical effort to keep control. The small joints of the hand, as well as the tendons can be strained by maintaining a firm grip on the handlebar in the same position for an extended period. Furthermore, the occurrence of hand vibrations while riding a motorbike [4], repetitive clutch holding, repeated use of hand breaks, and the task of carrying passengers all contribute to the heightened challenges faced by motorcyclists, potentially increasing the likelihood of developing hand syndrome. The highest prevalence of pain in the right hand was reported in area F (the palm and near the wrist), which may be linked to prolonged pressure on the handlebar while using the throttle. In contrast, a significant prevalence of pain was observed in area D of the left hand, likely associated with repetitive use of the clutch. These findings highlight the importance of motorcycle handlebar ergonomics.

This present study found significant differences in education, marital status, and rider status. M. Ali et al. reported a significant association between low educational attainment and low back pain among bike riders in Bangladesh. Our present study also found participants with lower educational qualifications, (HSC and SSC or below) exhibited a higher susceptibility to hand syndrome compared to those with higher educational qualifications (bachelor's degree and above) which is similar to the findings of the study conducted by M. Ali et al. The study also reported that marital status and rider

**Table 3. Binary logistic regression analysis: predictors of biker's hand syndrome.**

| Variables | Right hand | | | Left hand | | |
|---|---|---|---|---|---|---|
| | AOR | 95% CI | p value | AOR | 95% CI | p value |
| **Age** | | | | | | |
| >40 years | reference | | | reference | | |
| 31-40 years | 0.267 | 0.113–0.534 | <0.001 | 0.332 | 0.172–0.637 | 0.001 |
| 18-30 years | 0.551 | 0.301–1.007 | 0.053 | 0.612 | 0.349–1.074 | 0.087 |
| **BMI** | | | | | | |
| ≥ 25 kg/m$^2$ | reference | | | reference | | |
| 23.0-24.9 kg/m$^2$ | 0.811 | 0.165–3.992 | 0.797 | 0.595 | 0.111–3.198 | 0.545 |
| 18.5-22.9 kg/m$^2$ | 2.010 | 1.238–3.263 | 0.005 | 1.628 | 1.015–2.610 | 0.043 |
| <18.5 kg/m$^2$ | 1.162 | 0.681–1.985 | 0.582 | 1.583 | 0.940 - 2.667 | 0.084 |
| **Educational Qualification** | | | | | | |
| Graduation and above | reference | | | reference | | |
| HSC | 2.929 | 1.558–5.506 | 0.001 | 1.791 | 0.982 −3.268 | 0.057 |
| ≤ SSC | 1.867 | 1.188–2.933 | 0.007 | 1.001 | 0.644 - 1.557 | 0.995 |
| **Marital status** | | | | | | |
| Married | reference | | 0.477 | reference | | 0.374 |
| Unmarried/Divorced/Widow | 1.223 | 0.728 - 2.054 | | 1.261 | 0.756–2.103 | |
| **Gross monthly income** | | | | | | |
| ≤ 30000 | reference | | | reference | | |
| 31000-45000 | 0.766 | 0.312 - 1.882 | 0.562 | 0.432 | 0.185–1.011 | 0.053 |
| >45000 | 1.223 | 0.456 - 2.677 | 0.825 | 0.791 | 0.347 - 1.803 | 0.578 |
| **Rider status** | | | | | | |
| Occasional | reference | | 0.015 | reference | | 0.001 |
| Professional | 2.208 | 1.168–4.171 | | 3.016 | 1.573 - 5.782 | |
| **Average riding time in a day** | | | | | | |
| > 8 hours | reference | | | reference | | |
| 6-8 Hours | 0.345 | 0.166 - 0.718 | 0.004 | 0.437 | 0.218 - 0.875 | 0.020 |
| < 6 hours | 0.557 | 0.283 - 1.096 | 0.090 | 0.760 | 0.403 - 1.431 | 0.395 |
| **Experience as a bike rider** | | | | | | |
| >10 years | reference | | | reference | | |
| 6-10 years | 0.844 | 0.339 - 2.098 | 0.714 | 1.160 | 0.508 - 2.652 | 0.724 |
| 0-5 years | 2.062 | 0.823 - 5.167 | 0.112 | 3.472 | 1.511–7.977 | 0.003 |
| **Bike engine capacity** | | | | | | |
| > 150 cc | reference | | | Reference | | |
| ≤ 150 cc | 2.218 | 1.192 - 4.128 | 0.012 | 1.210 | 0.672 - 2.157 | 0.525 |
| **Is the hand grip of your bike suitable for you?** | | | | | | |
| Yes | reference | | 0.013 | reference | | 0.127 |
| No | 2.110 | 1.171 - 3.801 | | 1.519 | 0.888 - 2.598 | |
| **Do you think your bike fits according to your body?** | | | | | | |
| Yes | reference | | <0.001 | reference | | <0.001 |
| No | 5.136 | 2.939 - 8.974 | | 3.676 | 2.210–6.113 | |

Dependent variable: presence of pain, ache or discomfort in either hand in the past 7 days, assessed by CHDQ, BMI – body mass index; SSC – secondary school certificate; HSC – higher secondary school certificate; AOR – adjusted odd ratio; CI – confidence interval; cc – cubic capacity

status were significantly associated with low back pain [5], which is similar to the result of this present study. Additionally, this study found no association between hand syndrome and smoking, and the use of safety equipment may indicate that the lack of a protective effect from safety equipment reflects either actual ineffectiveness or improper use.

The study's participants exhibit a range of hand syndromes throughout various shaded regions of their right and left hands. 58.3% of participants indicated experiencing overall discomfort in their right hand, with 20.5% reporting pain in region A, 12.7% in area B, 16.5% in area C, 13.0% in area D, 16.8% in area E, and 22.2% in area F. Conversely, 51.3% of participants indicated experiencing overall discomfort in their left hand, with 16.2% reporting pain in region A, 10.5% in area B, 10.2% in area C, 16.7% in area D, 10.5% in area E, and 14.3% in area F. This study exhibits a difference in hand syndrome between right and left hand also. Motorcyclists in Bangladesh utilize the right hand for braking and throttling, while the left hand operates the clutch, necessitating distinct muscular engagement for each function. The disparity in hand syndrome may result from the utilization of distinct muscle patterns for various tasks. In our study, individuals who ride a bike with an engine capacity of less than 150cc are twice as likely to develop hand syndrome. A recent published study by M. Ali et al. observed similar results, reporting that riding a bike with less cubic capacity provoked low back pain [5]. On the other hand, riding a high-powered motorcycle may provoke hand-arm vibration syndrome among male police-men [13].

### Strength, limitations and future recommendations

In this cross-sectional study, we used face-to-face interviewer-administered questionnaires, which helped reduce the likelihood of non-response and misclassification bias, resulting in more precise and reliable data. Employing a two-stage cluster random sample method from two separate metropolitan areas improves result precision and reduces selection bias. Including the entire nation's ride-sharing riders could increase the generalizability of the findings. This study has certain limitations that need to be acknowledged. Due to its cross-sectional nature, this study is unable to establish a causal relationship between the independent and dependent variables. Furthermore, it should be noted that the results of this study may vary in different urban or rural regions of Bangladesh, as the research was particularly carried out in two metro-politan areas where only professional bikers were included. Variations in road infrastructure, traffic congestion, motorbike types, or work habits may influence hand pain. In Dhaka, the elevated traffic congestion and poor road conditions, relative to other cities, may augment biomechanical stress on the hands. One of our limitations that was not addressed in this study was the road pavement conditions. Traffic in Dhaka Metropolitan City is somewhat higher than in other metropolitan cities in Bangladesh. The participants' hand syndrome may also be influenced by the traffic condition and road mainte-nance status. This study employed the CHDQ, which has not been validated for motorcycle riders or in the context of Bangladesh, potentially compromising the credibility of the findings. The reliance on self-reported questionnaires hinders our ability to distinguish BHS from other symptoms and to understand the connections with other co-morbidities, such as diabetes, arthritis, or a history of repetitive hand use, due to the absence of clinical or neurological investigation. Finally, the cross-sectional design of the current study suggests a need for future longitudinal research to examine symptom prog-nosis. Additionally, intervention studies focusing on ergonomic modifications, such as handlebar design and glove usage, should be conducted to assess their impact on hand pain syndrome. Furthermore, due to the unmeasured influence of road conditions, comparative studies that include non-professional riders or other transportation workers could help iden-tify risk factors that are specific to motorcycle riding.

### Conclusion

Professional bikers in Bangladesh have reported experiencing hand syndrome in multiple regions of their hands. Individ-uals with the age group less than 40 had a lower prevalence of hand syndrome than those over 40. Moreover, persons who are underweight have an elevated susceptibility to acquiring hand syndrome. Individuals who ride motorcycles with an engine capacity of less than 150cc had a risk of developing hand syndrome that was twice as high as those who ride

motorcycles with an engine capacity of more than 150cc. Bikers should use comfortable bike fitted according to their body physique and should use comfortable hand gripper to avoid pain. Individuals whose motorcycles do not match their body physique have more than five times the likelihood of experiencing hand discomfort. It is strongly advised that ride-sharing platforms implement policies for motorcycle selection or adjustments based on the rider's height and weight.

## Key-points:

- Motorbike ride-sharing is becoming increasingly popular worldwide and is a significant cause of occupational injuries. Prior studies primarily concentrated on musculoskeletal problems, specifically low back pain, carpal tunnel syndrome, and other similar conditions. Nevertheless, there is a scarcity of comprehensive research on musculoskeletal injuries in the hand, which limits people's understanding of disease prevention and treatment.

- This study conducted a questionnaire survey on bikers from two different metropolitan cities in Bangladesh and found that more than half of the participants had hand syndrome. Factors like age, BMI, experience, the bike's lower engine capacity, the suitability of the hand grip, and the bike's alignment with the body's physique all play a significant role.

- Our research findings will provide research evidence for the prevention and control of hand syndrome, as well as basic information for the formulation of relevant policies to develop more effective health promotion measures and work environment improvement plans.

## Supporting information

**S1 Text. STROBE Checklist.**
(DOCX)

**S2 Text. Questionnaire.**
(DOCX)

**S3 Data. Data – Basic life Support.**
(XLSX)

## Acknowledgments

We thank all the participants involved in this study.

## Author contributions

**Conceptualization:** Sohel Ahmed, Mohammad Jahirul Islam, G M Jakaria, Md. Enamul Haque, Jalal Uddin, Tazveen Fariha, Md Saifur Rahman, Md. Zahidul Islam, Raju Ahmed, Selim Hossain, S. M. Mahfuz Anwar.

**Data curation:** Sohel Ahmed, Mohammad Jahirul Islam, G M Jakaria, Md. Enamul Haque, Jalal Uddin, Tazveen Fariha, Md Saifur Rahman, Md. Zahidul Islam, Raju Ahmed, Selim Hossain.

**Formal analysis:** Sohel Ahmed.

**Investigation:** Sohel Ahmed.

**Methodology:** Sohel Ahmed, Mohammad Jahirul Islam, G M Jakaria, Md. Enamul Haque, Jalal Uddin, Tazveen Fariha, Md Saifur Rahman, Md. Zahidul Islam, Raju Ahmed, Selim Hossain, S. M. Mahfuz Anwar.

**Project administration:** Sohel Ahmed.

**Software:** Sohel Ahmed.

**Supervision:** S. M. Mahfuz Anwar.

**Validation:** Sohel Ahmed.

**Visualization:** Sohel Ahmed.

**Writing – original draft:** Sohel Ahmed, Mohammad Jahirul Islam, G M Jakaria, Md. Enamul Haque, Jalal Uddin, Tazveen Fariha, Md Saifur Rahman.

**Writing – review & editing:** Sohel Ahmed, S. M. Mahfuz Anwar.

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
