## [Decision Letter · Decision Letter 0]

7 Aug 2025

PONE-D-25-11037Prevalence and predictors of Biker’s hand syndrome among the professional bike riders in Bangladesh: A cross-sectional studyPLOS ONE

Dear Dr. Sohel Ahmed,

Thank you for submitting your manuscript to PLOS ONE. After careful consideration, we feel that it has merit but does not fully meet PLOS ONE’s publication criteria as it currently stands. Therefore, we invite you to submit a revised version of the manuscript that addresses the points raised during the review process.

We look forward to receiving your revised manuscript.

Kind regards,

Nik Hisamuddin Nik Ab. Rahman

Academic Editor

PLOS ONE

Reviewers' comments:

Reviewer's Responses to Questions

**Comments to the Author**

1. Is the manuscript technically sound, and do the data support the conclusions?

Reviewer #1: Partly

Reviewer #2: Partly

2. Has the statistical analysis been performed appropriately and rigorously? 

Reviewer #1: Yes

Reviewer #2: Yes

3. Have the authors made all data underlying the findings in their manuscript fully available?

Reviewer #1: Yes

Reviewer #2: Yes

4. Is the manuscript presented in an intelligible fashion and written in standard English?

Reviewer #1: Yes

Reviewer #2: Yes

5. Review Comments to the Author

Reviewer #1: This cross-sectional study investigates the prevalence and predictors of Biker’s Hand Syndrome (BHS) among 630 professional ride-share motorcyclists in Bangladesh. Using the Cornell Hand Discomfort Questionnaire, it reports a high prevalence of hand discomfort (58.3% right, 51.3% left) and identifies significant associations with age, BMI, years of riding, engine capacity, grip suitability, and bike-body fit. The study is methodologically sound, with appropriate sampling and statistical modeling. It addresses a research gap by focusing on hand-specific musculoskeletal disorders in a developing country occupational setting. Its practical implications are clear—highlighting ergonomic risk factors that could inform health policy, rider training, and bike design. However, the reliance on self-reported symptoms without clinical validation limits diagnostic certainty, and factors like comorbidities or road conditions were not explored. Despite these limitations, the findings offer valuable direction for ergonomic interventions and occupational health strategies in the ride-sharing sector.

A few major issues need to be addressed before publication.

1. Lack of Clinical Validation of Biker’s Hand Syndrome: While the authors acknowledge using a self-reported questionnaire, they do not explicitly discuss the limitation of lacking clinical or neurological confirmation (e.g., no physical or diagnostic assessment to differentiate BHS from other hand disorders). This remains a significant methodological limitation.

2. Omission of Confounding Variables: The study does not mention or control for comorbidities such as diabetes, arthritis, or history of repetitive hand use, which could influence symptom reporting. These are important omissions that were not addressed in the limitations.

3. No Justification for Right vs. Left Hand Modeling Strategy: Although right and left hand data are reported separately, the rationale for modeling them independently is not discussed, and inter-hand correlation is not addressed. This may affect interpretation of results.

A few minor issues need to be addressed before publication.

1. Terminology and Clarity: “Biker’s Hand Syndrome” is treated as a distinct condition without formal clinical recognition. A brief clarification of the term’s novelty or how it relates to established conditions (e.g., hand-arm vibration syndrome) would improve clarity.

2. Formatting and Typographical Issues: Minor inconsistencies remain (e.g., “echo friendly” instead of “eco-friendly”), but these are easily corrected.

3. Limited Depth on Preventive Strategies: Although prevention is mentioned (e.g., bike fit, stretching), these suggestions are not supported with specific references or actionable guidelines, making the recommendations feel superficial.

4. Lack of Visual Summary: A flowchart of the sampling and data analysis procedure or a forest plot summarizing regression findings would improve accessibility, especially for visual readers.

5. CHDQ Validation Context: The CHDQ is cited, but its cultural adaptation and validation for Bangladeshi riders is not explained. This should be clarified to support its use in this specific context.

Formatting and spelling issues:

1. Line number: Please add line number to help the reviewer to read and locate the issues

2. echo friendly (pdf p.15, article p.9, Survey administration section): should be eco-friendly

3. hand gripper that provides comfort (abstract and conclusion): it would be better “comfortable hand gripper”

4. MBI (pdf p.20, article p.14, Discussion): BMI

5. affirmative or negative response (pdf p. 12, article p.6, Ethical consideration section): it would be better “consent or decline”

Reviewer #2: Review Comments to the Author

The manuscript presents a timely and relevant study on the prevalence and predictors of biker’s hand syndrome among professional motorcycle riders in Bangladesh—a growing occupational group facing significant musculoskeletal challenges. The topic is important from both public health and ergonomic perspectives, particularly in low- and middle-income countries where ride-sharing services are expanding rapidly. The study design, data collection methods, and statistical analyses are generally sound, and the findings offer valuable insights into a largely understudied occupational health issue. However, several revisions are needed to improve clarity, methodological transparency, and scientific rigor before the manuscript can be considered for publication.

The Introduction effectively establishes the context of motorcycle use in Bangladesh and defines biker’s hand syndrome with reference to pain, discomfort, and functional impact. The justification for the study is reasonable, highlighting a gap in the literature regarding professional riders. However, the claim that “no study has yet been conducted on this serious issue” is likely an overstatement. While the specific focus on ride-sharing riders in Bangladesh may be novel, similar conditions have been studied in cyclists and motorcyclists elsewhere. A more precise statement—such as “no prior study has specifically investigated hand discomfort among professional ride-sharing riders in Bangladesh”—would be more accurate and defensible. Additionally, the introduction would benefit from a clearer conceptual framework linking individual, occupational, and environmental factors to the development of hand syndrome.

The Methods section is generally well-structured, but several improvements are needed. First, the definition of “professional biker” requires clarification. Does this term refer exclusively to ride-sharing drivers (e.g., Pathao, Uber), or does it include other types of motorcycle-based workers? A clear operational definition (e.g., income source, daily riding hours) should be provided. Second, while the use of two-stage cluster sampling is appropriate, the process of randomization is not sufficiently detailed. Please specify how clusters (e.g., bus stations, markets) were selected—was a sampling frame used, or was selection based on convenience? Third, although the Cornell Hand Discomfort Questionnaire (CHDQ) was used, only the presence/absence of pain was reported, not severity or frequency. This limits the depth of analysis. Please justify this decision and clarify whether the full CHDQ was administered. Additionally, while back-translation was performed, there is no mention of pilot testing results or reliability assessment (e.g., Cronbach’s alpha) in the target population, which is essential for validating translated instruments.

The Results section is comprehensive and well-presented, with clear reporting of descriptive statistics, prevalence rates, bivariate associations, and multivariable regression models. The separate analysis of right and left hand symptoms is clinically meaningful. However, several issues require attention. First, the dependent variable in the logistic regression—“presence of pain in any of six hand regions”—should be explicitly defined in the text or in a table footnote. Second, while adjusted odds ratios (aOR) and 95% CIs are reported, p-values for each predictor are missing. These should be included for full transparency. Third, model fit statistics (e.g., Hosmer-Lemeshow test result, p-value) and multicollinearity diagnostics (e.g., VIF values) should be reported to support the validity of the regression model. Fourth, the finding that normal BMI is associated with higher odds of hand syndrome compared to obesity is counterintuitive and requires careful interpretation in the Discussion—possibly due to confounding or differential exposure patterns.

The Strengths, Limitations, and Future Recommendations section acknowledges key limitations, including the cross-sectional design and inability to infer causality, which is commendable. However, the claim that the findings are generalizable to the “entire nation’s populace” is inaccurate and should be revised. The study was conducted in only two metropolitan areas and among a specific occupational group, limiting broader applicability. Instead, the authors should emphasize urban generalizability and suggest future studies in rural or semi-urban settings. The omission of road surface quality and traffic conditions as measured variables is a valid limitation, as these factors significantly influence hand vibration and strain. Future studies should consider incorporating objective or self-reported measures of these environmental exposures.

Finally, the future research recommendations are somewhat generic. To enhance impact, they should be more specific and actionable—such as recommending longitudinal designs to assess symptom progression, or intervention studies evaluating ergonomic modifications (e.g., handlebar design, glove use). Comparative studies with non-professional riders or other transport workers could also deepen understanding of occupational risk factors.

Minor issues include repetitive phrasing, inconsistent use of terms (e.g., “pain, ache, or discomfort”), and lack of a figure showing the CHDQ hand map (regions A–F). Including such a figure would greatly improve clarity. Additionally, ensure all statistical software (e.g., SPSS, R) and version numbers are reported.

In summary, this is a valuable contribution to occupational health research with strong potential. With the suggested revisions—particularly in clarifying methods, improving statistical reporting, refining interpretations, and enhancing future directions—the manuscript will be well-suited for publication in an international journal.

6. PLOS authors have the option to publish the peer review history of their article (what does this mean? ). If published, this will include your full peer review and any attached files.

**Do you want your identity to be public for this peer review?** For information about this choice, including consent withdrawal, please see our Privacy Policy .

Reviewer #1: **Yes: ** Pu-Chun Mo

Reviewer #2: No

---

## [Author Response · Author response to Decision Letter 1]

24 Aug 2025

Reviewer #1

This cross-sectional study investigates the prevalence and predictors of Biker’s Hand Syndrome (BHS) among 630 professional ride-share motorcyclists in Bangladesh. Using the Cornell Hand Discomfort Questionnaire, it reports a high prevalence of hand discomfort (58.3% right, 51.3% left) and identifies significant associations with age, BMI, years of riding, engine capacity, grip suitability, and bike-body fit. The study is methodologically sound, with appropriate sampling and statistical modeling. It addresses a research gap by focusing on hand-specific musculoskeletal disorders in a developing country occupational setting. Its practical implications are clear—highlighting ergonomic risk factors that could inform health policy, rider training, and bike design. However, the reliance on self-reported symptoms without clinical validation limits diagnostic certainty, and factors like comorbidities or road conditions were not explored. Despite these limitations, the findings offer valuable direction for ergonomic interventions and occupational health strategies in the ride-sharing sector.

Reviewer’s comment: 1. Lack of Clinical Validation of Biker’s Hand Syndrome: While the authors acknowledge using a self-reported questionnaire, they do not explicitly discuss the limitation of lacking clinical or neurological confirmation (e.g., no physical or diagnostic assessment to differentiate BHS from other hand disorders). This remains a significant methodological limitation.

Reply to the reviewer’s comment: Thank you for your valuable comment and for highlighting this important issue. We have now addressed it in the discussion section as a limitation in the revised text.

Reviewer’s comment: 2. Omission of Confounding Variables: The study does not mention or control for comorbidities such as diabetes, arthritis, or history of repetitive hand use, which could influence symptom reporting. These are important omissions that were not addressed in the limitations.

Reply to the reviewer’s comment: Thank you for your insightful remark and for emphasizing this significant matter. The issue has been resolved in the discussion section as a limitation in the amended text.

Reviewer’s comment: 3. No Justification for Right vs. Left Hand Modeling Strategy: Although right- and left-hand data are reported separately, the rationale for modeling them independently is not discussed, and inter-hand correlation is not addressed. This may affect interpretation of results.

Reply to the reviewer’s comment: Thank you for bringing up this issue. We aim to investigate the prevalence and predictors of hand syndrome among professional ride-sharing riders in Bangladesh. When riding a motorcycle, the right and left hands engage differently with the clutch, throttle, and hand brakes, necessitating the use of both hands. For that reason, we are modeling them independently. For your clarification, we now added the following text in the revised manuscript: ‘’The highest prevalence of pain in the right hand was reported in area F (the palm and near the wrist), which may be linked to prolonged pressure on the handlebar while using the throttle. In contrast, a significant prevalence of pain was observed in area D of the left hand, likely associated with repetitive use of the clutch. These findings highlight the importance of motorcycle handlebar ergonomics.’’

Reviewer’s comment: 1. Terminology and Clarity: “Biker’s Hand Syndrome” is treated as a distinct condition without formal clinical recognition. A brief clarification of the term’s novelty or how it relates to established conditions (e.g., hand-arm vibration syndrome) would improve clarity.

Reply to the reviewer’s comment: Thank you for your valuable comment, we already defined the terminology in the introduction section of the manuscript as follows: "Biker's hand syndrome is a condition characterized by pain, discomfort, paresthesia, or numbness in the hands. The symptoms result from excessive usage, nerve compression, or excessive pressure on the hand during prolonged durations of bike riding.’’

Reviewer’s comment: 2. Formatting and Typographical Issues: Minor inconsistencies remain (e.g., “echo friendly” instead of “eco-friendly”), but these are easily corrected.

Reply to the reviewer’s comment: Thank you, corrected in the revised text.

Reviewer’s comment: 3. Limited Depth on Preventive Strategies: Although prevention is mentioned (e.g., bike fit, stretching), these suggestions are not supported with specific references or actionable guidelines, making the recommendations feel superficial.

Reply to the reviewer’s comment: Thank you for your valuable comment. We now addressed the following statements in the revised text: ‘’Bikers should use comfortable bike fitted according to their body physique and should use comfortable hand gripper to avoid pain. Individuals whose motorcycles do not match their body physique have more than five times the likelihood of experiencing hand discomfort. It is strongly advised that ride-sharing platforms implement policies for motorcycle selection or adjustments based on the rider’s height and weight.’’

Reviewer’s comment: 4. Lack of Visual Summary: A flowchart of the sampling and data analysis procedure or a forest plot summarizing regression findings would improve accessibility, especially for visual readers.

Reply to the reviewer’s comment: Thank you for your valuable suggestion. The revised text now includes Figure 2, which provides details on the sampling and data analysis.

Reviewer’s comment: 5. CHDQ Validation Context: The CHDQ is cited, but its cultural adaptation and validation for Bangladeshi riders is not explained. This should be clarified to support its use in this specific context.

Reply to the reviewer’s comment: The cross-cultural adaptation or language validation for this questionnaire in Bangla has not yet been conducted. However, to ensure consistency, we employed forward and backward translation for the questionnaire, and we trained all data collectors before the data collection process.

Reviewer’s comment: 1. Line number: Please add line number to help the reviewer to read and locate the issues

Reply to the reviewer’s comment: added in the revised text.

Reviewer’s comment: echo friendly (pdf p.15, article p.9, Survey administration section): should be eco-friendly

Reply to the reviewer’s comment: Thank you for highlighting our typographical mistake. Corrected in the revised text.

Reviewer’s comment: hand gripper that provides comfort (abstract and conclusion): it would be better “comfortable hand gripper”

Reply to the reviewer’s comment: Thank you for your comment. Corrected in the revised text.

Reviewer’s comment: MBI (pdf p.20, article p.14, Discussion): BMI

Reply to the reviewer’s comment: Thank you for highlighting our typographical mistake. Corrected in the revised text.

Reviewer’s comment: affirmative or negative response (pdf p. 12, article p.6, Ethical consideration section): it would be better “consent or decline”

Reply to the reviewer’s comment: Corrected in the revised text.

Reviewer #2

The manuscript presents a timely and relevant study on the prevalence and predictors of biker’s hand syndrome among professional motorcycle riders in Bangladesh—a growing occupational group facing significant musculoskeletal challenges. The topic is important from both public health and ergonomic perspectives, particularly in low- and middle-income countries where ride-sharing services are expanding rapidly. The study design, data collection methods, and statistical analyses are generally sound, and the findings offer valuable insights into a largely understudied occupational health issue. However, several revisions are needed to improve clarity, methodological transparency, and scientific rigor before the manuscript can be considered for publication.

Reviewer’s comment: The Introduction effectively establishes the context of motorcycle use in Bangladesh and defines biker’s hand syndrome with reference to pain, discomfort, and functional impact. The justification for the study is reasonable, highlighting a gap in the literature regarding professional riders. However, the claim that “no study has yet been conducted on this serious issue” is likely an overstatement. While the specific focus on ride-sharing riders in Bangladesh may be novel, similar conditions have been studied in cyclists and motorcyclists elsewhere. A more precise statement—such as “no prior study has specifically investigated hand discomfort among professional ride-sharing riders in Bangladesh”—would be more accurate and defensible. Additionally, the introduction would benefit from a clearer conceptual framework linking individual, occupational, and environmental factors to the development of hand syndrome.

Reply to the reviewer’s comment: Thank you for your comments. As per your recommendation we now replace the statement “no study has yet been conducted on this serious issue” with “no prior study has specifically investigated hand discomfort among professional ride-sharing riders in Bangladesh” in the revised text. We also included a conceptual framework in the revised version of the manuscript.

Reviewer’s comment: The Methods section is generally well-structured, but several improvements are needed. First, the definition of “professional biker” requires clarification. Does this term refer exclusively to ride-sharing drivers (e.g., Pathao, Uber), or does it include other types of motorcycle-based workers? A clear operational definition (e.g., income source, daily riding hours) should be provided. Second, while the use of two-stage cluster sampling is appropriate, the process of randomization is not sufficiently detailed. Please specify how clusters (e.g., bus stations, markets) were selected—was a sampling frame used, or was selection based on convenience? Third, although the Cornell Hand Discomfort Questionnaire (CHDQ) was used, only the presence/absence of pain was reported, not severity or frequency. This limits the depth of analysis. Please justify this decision and clarify whether the full CHDQ was administered. Additionally, while back-translation was performed, there is no mention of pilot testing results or reliability assessment (e.g., Cronbach’s alpha) in the target population, which is essential for validating translated instruments.

Reply to the reviewer’s comment: Thank you for your valuable comments. The revised manuscript includes the following statement to clarify who is a professional biker:

The term "professional biker" specifically refers to ride-sharing drivers (such as Pathao and Uber) who generate income by providing ride-sharing services and who ride bikes for at least 3 to 4 hours each day. The cluster sampling process is now described in the revised text.

Thirdly, the objective of this study was to determine the prevalence and predictors of biker's hand syndrome among professional bike riders in Bangladesh. We reported the frequency, severity, and interference in Table 1; however, in line with our objective, we focused solely on the presence or absence of pain (prevalence) in the shaded area and analyzed the data accordingly.

To ensure data collectors were competent and to identify any inconsistencies during data collection, we conducted a pilot test of the questionnaire with five participants for each data collector. We acknowledge a limitation in that we did not assess the reliability of the pilot study, as noted in the limitations section of the revised manuscript.

Reviewer’s comment: The Results section is comprehensive and well-presented, with clear reporting of descriptive statistics, prevalence rates, bivariate associations, and multivariable regression models. The separate analysis of right- and left-hand symptoms is clinically meaningful. However, several issues require attention. First, the dependent variable in the logistic regression— “presence of pain in any of six hand regions”—should be explicitly defined in the text or in a table footnote. Second, while adjusted odds ratios (aOR) and 95% CIs are reported, p-values for each predictor are missing. These should be included for full transparency. Third, model fit statistics (e.g., Hosmer-Lemeshow test result, p-value) and multicollinearity diagnostics (e.g., VIF values) should be reported to support the validity of the regression model. Fourth, the finding that normal BMI is associated with higher odds of hand syndrome compared to obesity is counterintuitive and requires careful interpretation in the Discussion—possibly due to confounding or differential exposure patterns.

Reply to the reviewer’s comment: Thank you for your comment. The dependent variable, which refers to the presence or absence of pain in the six shaded areas, is already included in Table 1. The p-value is mentioned in Table 4; however, we have also included it in the text of the revised manuscript. The model fitness statistics also added in the data analysis section.

Reviewer’s comment: The Strengths, Limitations, and Future Recommendations section acknowledges key limitations, including the cross-sectional design and inability to infer causality, which is commendable. However, the claim that the findings are generalizable to the “entire nation’s populace” is inaccurate and should be revised. The study was conducted in only two metropolitan areas and among a specific occupational group, limiting broader applicability. Instead, the authors should emphasize urban generalizability and suggest future studies in rural or semi-urban settings. The omission of road surface quality and traffic conditions as measured variables is a valid limitation, as these factors significantly influence hand vibration and strain. Future studies should consider incorporating objective or self-reported measures of these environmental exposures.

Reply to the reviewer’s comment: We now corrected the generalizability statement as follow in the revised text: ‘’Employing a random sample method from two separate metropolitan areas improves result precision and reduces selection bias. Encompassing the entire nation's populace could increase the generalizability of the findings.’’

Reviewer’s comment: Finally, the future research recommendations are somewhat generic. To enhance impact, they should be more specific and actionable—such as recommending longitudinal designs to assess symptom progression, or intervention studies evaluating ergonomic modifications (e.g., handlebar design, glove use). Comparative studies with non-professional riders or other transport workers could also deepen understanding of occupational risk factors.

Reply to the reviewer’s comment: Thank you for your insightful comment. The future recommendation section is now upgraded, as per your suggestion in the revised text.

Reviewer’s comment: Minor issues include repetitive phrasing, inconsistent use of terms (e.g., “pain, ache, or discomfort”), and lack of a figure showing the CHDQ hand map (regions A–F). Including such a figure would greatly improve clarity. Additionally, ensure all statistical software (e.g., SPSS, R) and version numbers are reported.

Reply to the reviewer’s comment: We already included figure 1, which includes a hand map diagram (regions A-F). We also reported the statistical software name and version in the revised text.

Reviewer’s comment: In summary, this is a valuable contribution to occupational health research with strong potential. With the suggested revisions—particularly in clarifying methods, improving statistical reporting, refining interpretations, and enhancing future directions—the manuscript will be well-suited for publication in an international journal.

Abstract

The abstract is well-structured and communicates key findings effectively. However, with the suggested revisions—particularly in enhancing methodological transparency, statistical reporting, interpretative clarity, and strategic use of keywords—it can become a stronger, more impactful representation of the study, significantly improving its chances of acceptance and readership in a high-quality journal.

Reviewer’s comment: The background section provides a general introduction to the topic but could be strengthened with more specific contextual details. The phrase "gaining popularity" is too vague; it w

---

## [Editor Report · Decision Letter 1]

28 Aug 2025

Prevalence and predictors of Biker’s hand syndrome among the professional bike riders in Bangladesh: A cross-sectional study

PONE-D-25-11037R1

Dear Dr. Ahmed,

We’re pleased to inform you that your manuscript has been judged scientifically suitable for publication and will be formally accepted for publication once it meets all outstanding technical requirements.

Kind regards,

Nik Hisamuddin Nik Ab. Rahman

Academic Editor

PLOS ONE
---

## [Editor Report · Acceptance letter]

PONE-D-25-11037R1

PLOS ONE

Dear Dr. Ahmed,

I'm pleased to inform you that your manuscript has been deemed suitable for publication in PLOS ONE. Congratulations! Your manuscript is now being handed over to our production team.

Kind regards,

on behalf of

Professor Dr Nik Hisamuddin Nik Ab. Rahman

Academic Editor

PLOS ONE